# Effects of Treatment Combining 9300 nm Carbon Dioxide Lasers and Fluoride on Prevention of Enamel Caries: A Systematic Review and Meta-Analysis

**Vicky Wenqing Xue** [1,2], **Irene Shuping Zhao** [2,3,*], **Iris Xiaoxue Yin** [1,2], **John Yun Niu** [1,2], **Edward Chin Man Lo** [1,2] and **Chun Hung Chu** [1,2,*]

1   Faculty of Dentistry, The University of Hong Kong, Hong Kong 999077, China; vickyxue@hku.hk (V.W.X.); irisxyin@hku.hk (I.X.Y.); niuyun1@hku.hk (J.Y.N.); edward-lo@hku.hk (E.C.M.L.)
2   HKU Shenzhen Institute of Research and Innovation, Shenzhen 518000, China
3   School of Dentistry, Shenzhen University Health Science Center, Shenzhen 518000, China
*   Correspondence: zhao110@szu.edu.cn (I.S.Z.); chchu@hku.hk (C.H.C.)

**Abstract:** The objective of this study is to conduct a systematic review of the literatures on the effect of treatment combining 9300 nm carbon dioxide ($CO_2$) lasers and fluoride on prevention of enamel caries. A literature search was performed using PubMed, Scopus, and Web of Science. The keywords were ((9300 nm) OR (9.3 μm) OR (carbon dioxide laser) OR (carbon dioxide lasers) OR ($CO_2$ laser) OR ($CO_2$ lasers)) AND ((fluoride) OR (fluorides)) AND ((dental caries) OR (caries) OR (remineralization) OR (remineralization) OR (demineralization) OR (demineralization)). Meta-analysis was performed to compare the change in mineral content by laser irradiation and fluoride treatment (F + L) with that of fluoride treatment (F) and laser treatment (L). The search identified 946 potential publications and five laboratory studies using a chemical model for cariogenic challenge and determining mineral loss of the enamel were included in this review. Meta-analysis comparing F + L with L on enamel showed a standard mean difference of −1.58 (CI: −2.13, −1.03). Meta-analysis comparing F + L with F on enamel showed a standard mean difference of −1.84 (CI: −2.30, −1.39), with low heterogeneity ($I^2 = 49\%$, $p = 0.04$). In conclusion, F + L was better than L and F in preventing enamel demineralization.

**Keywords:** fluoride; caries; prevention; laser; enamel; demineralization; remineralization; carbon dioxide

## 1. Introduction

Dental caries are a major global health problem [1]. They lead to the demineralization and proteolytic destruction of the organic component of dental tissues as a progressively infectious disease that ultimately causes the formation of cavities [2]. The metabolic actions of oral microorganisms upon dietary carbohydrates in the plaque biofilm produce acid and cause mineral loss in enamel and dentine [3]. Many clinical and laboratory studies have investigated a variety of methods to reduce mineral loss in hard dental tissues against cariogenic challenge. Among them, the application of topical fluoride in the forms of gel, varnish, toothpaste, or mouth rinse has been extensively used as a benchmark intervention for the prevention of dental caries [4].

Currently, the most commonly used topical fluoride agents for individual or professional application are sodium fluoride and acidulated phosphate fluoride with various concentrations. The topical application of fluoride reduces the demineralization process, and it possesses a more profound remineralizing effect [5]. Fluoride can unite with calcium ions and hydrogen phosphate ions to be incorporated physiochemically into the hydroxyapatite lattice structure, forming fluorapatite and fluorhydroxyapatite which are more resistant to acid [6]. Fluoride deposited on the tooth surface can produce calcium fluoride,

which acts as a temporary fluoride reservoir and can release fluoride ions in an acidic environment [7]. In addition, the topical use of fluoride can delay the progression of lesions and inhibit the formation of biofilm by binding to bacterial cellular components, and it can influence enzymes related to carbohydrate metabolism [8,9]. Fluoride also inhibits collagenases and thus hinders the degradation of dentine collagen [10].

Another strategy in preventing dental caries is the use of lasers [11]. Lasers can be absorbed by hydroxyapatite, the main mineral composition of dental enamel and dentine [12]. This absorption causes chemical and morphological changes such as the reduction of carbonate content and the recrystallization of hard dental tissues [11]. Compared with lasers at other wavelengths, such as argon lasers or neodymium-doped yttrium aluminum garnet (Nd:YAG) lasers, carbon dioxide ($CO_2$) lasers have a higher absorption coefficient to enamel. Therefore, carbon dioxide ($CO_2$) lasers may be a promising treatment for the prevention of caries [13]. $CO_2$ lasers have been found to increase the enamel's acid resistance to caries [14]. $CO_2$ lasers produce various emission laser lines with wavelengths ranging from 9000 nm to 11,000 nm [15]. Laser technology has evolved rapidly in recent years and has been proven as an effective strategy for the prevention of caries. A short-pulsed 9300 nm $CO_2$ laser is now commercially available and is one of the lasers most highly absorbed by the mineral components of hard dental tissue. This 9300 nm $CO_2$ laser has higher absorption but lower reflectance in hydroxyapatite than the conventional $CO_2$ laser (10,600 nm) [16]. Thus, it possesses a more effective energy transfer ability to dental hard tissues. A study showed that irradiation with a 9300 nm $CO_2$ laser can inhibit more than 60% of enamel caries compared with the negative control [17]. Moreover, because the 9300 nm $CO_2$ laser better matches the absorption characteristics of dental hard tissues, it can localize heat deposition to the surface and reduce the risks of thermal damage to the pulp [18,19]. Thus, a 9300 nm $CO_2$ laser can be used safely and effectively for the prevention of caries.

There is a general consensus that topically applied fluoride can reduce demineralization and enhance remineralization via physical and chemical interventions, however, its effect is partial and cannot prevent demineralization completely. The effect of topical fluoride on the prevention of caries depends on its constant presence in the oral cavity and also relies on the patient's oral hygiene [20]. Studies have indicated that laser treatment can work synergistically with the application of topical fluoride to further enhance its inhibitory effect [11,21]. Many studies have shown the effect of treatment combining irradiation by carbon dioxide laser at different wavelengths and topical fluoride application on inhibiting tooth demineralization [17,22–27]. A study demonstrated that the combined use of a $CO_2$ laser and sodium fluoride can significantly increase the uptake of fluoride ions [28]. Another study reported that the combined treatment resulted in the transformation of hydroxyapatite to fluorapatite, which is more resistant to acid dissolution [29]. A clinical trial reported that a treatment combining $CO_2$ laser irradiation and sodium-fluoride application significantly inhibited the formation of fissure caries in molars after 12 months [30]. Moreover, this combined treatment greatly enhanced the remineralization of enamel [31].

As the 9300 nm carbon dioxide laser has only been on the market for a relatively short period of time, there are far less studies using this equipment in comparison to the conventional 10,600 nm $CO_2$ laser. The result and the quality of these studies varies, and no consensus has been reached regarding the effect of the 9300 nm carbon dioxide laser and fluoride on the prevention of caries. As individual study cannot provide strong enough evidence to draw a generalized conclusion, we conducted a systematic review and meta-analysis of all the published studies in the literature. We aimed to provide clinicians and researchers with preliminary evidence. This systematic review and meta-analysis are the first to verify the pooled effect of data from studies that compare treatments combining fluoride application and 9300 nm $CO_2$ laser irradiation with a separate application for the prevention of caries. No similar review so far can be found in the literature.

## 2. Materials and Methods

This study was carried out in accordance with the preferred reporting items for systematic reviews [32], the meta-analyses (PRISMA) statement, and the guidelines of the Cochrane Handbook for the Systematic Reviews of Interventions [33].

### 2.1. Focused Question

Does enamel treated with a 9300 nm $CO_2$ laser and fluoride result in less mineral loss against cariogenic challenge than enamel treated by laser or fluoride separately?

### 2.2. Search Strategy

A literature search in English was performed by two independent investigators (Xue and Yin) to identify potentially relevant studies related to the research question across three electronic databases: MEDLINE via PubMed, the Web of Science core collection, and Scopus. There were no restrictions on the date of publication. The following key words were used to develop a list of potentially eligible studies based on a combination of specific medical subject headings (MeSH) and free text terms for these three databases: ((9300 nm) OR (9.3 µm) OR (carbon dioxide laser) OR (carbon dioxide lasers) OR ($CO_2$ laser) OR ($CO_2$ lasers)) AND ((fluoride) OR (fluorides)) AND ((dental caries) OR (caries) OR (remineralization) OR (remineralization) OR (demineralization) OR (demineralization)). The last search was performed on 31st January 2021 and studies published up to 31st December 2020 were included.

### 2.3. Study Selection and Data Extraction

Studies were selected for this review according to the following criteria for inclusion and exclusion.

#### 2.3.1. Inclusion Criteria

Original research investigating the effect of treatment combining 9300 nm $CO_2$ lasers and fluoride on the prevention of enamel caries were included in this systematic review. Any types of studies (including in vitro, in vivo, or clinical studies) related to fluoride applications and laser treatment modalities were included.

#### 2.3.2. Exclusion Criteria

1. Studies on other lasers or $CO_2$ lasers wavelengths other than 9300 nm;
2. Studies on dental erosion;
3. Literature reviews, case reports, conference papers, and book sections.

Duplicate literature was removed using the EndNote X7 software (Thomson Reuters, New York, NY, USA) and checked by two independent investigators (Xue and Yin). The investigators excluded literature reviews, book sessions, conference papers, and unrelated studies according to the inclusion and exclusion criteria via screening titles and abstracts. Afterwards, they retrieved the full texts of the remaining articles. The authors of unavailable literature were contacted for access to the full-text articles. Only studies that were available in full-text were considered for further screening. The reference lists of the identified full-text articles were also screened manually for relevant studies. Any disagreements on inclusion in the study were discussed by the two authors with a third author (Niu) until a consensus was reached.

### 2.4. Data Extraction

For the included studies, the following information was systematically recorded and checked by two independent investigators: publication details (title, authors, and years), methods (sample size, type of tooth, study design, and cariogenic challenge model), laser parameters used in studies, type and usage of fluoride, outcome assessments (various measurements for studying the prevention of caries), and outcome variables (mean values and standard deviations).

*2.5. Assessment of Risk of Bias*

Concerning the included in vitro studies, the risk of bias assessment was based on and adapted from the judgement model suggested by previous systematic reviews [34–36] for the assessment of risk of bias in in vitro trials. Therefore, the following eight domains were considered: quality check of samples; randomization of samples; sample-size calculation; homogeneity of samples; details of laser parameters used; fluoride application protocol; operator training; and blinding of operator. If the study met the criteria, the paper received a "yes" on that specific domain; if the relevant information could not be found, the paper received an "-". Two investigators (Xue and Yin) carried out the evaluation independently and inconsistencies were solved through discussion with a third author (Niu) until a consensus was reached. The risk of bias was classified according to the sum of "yes" answers received as follows: 1–3 = high, 4–5 = medium, 6–8 = low risk of bias.

*2.6. Data Analysis*

For each selected study, only the data including the sample size, mean, and standard deviation of mineral loss ($\Delta Z$) (vol% $\times$ mm) were extracted to be analyzed in the meta-analysis. For studies that evaluated several laser parameters, the single sample size, mean, and standard deviation for each condition were extracted. Considering the clear presence of heterogeneity between the methodologies used by the authors, the meta-analysis was conducted using a random-effect model based on the standardized mean difference in mineral-loss-related data to estimate the pooled effect of each intervention. Statistical significance was set at 0.05. Statistical heterogeneity among studies was assessed using the Cochrane Q statistic and $I^2$ test (0–40%: considered not important; 30% to 60%: may represent moderate heterogeneity; 50% to 90%: may represent substantial heterogeneity; 75% to 100%: represents considerable heterogeneity) [33]. When $I^2 > 50\%$, subgroups were created and a separate analysis was carried out considering the distinct measurement for evaluating the demineralization of caries (mineral loss measured by cross-sectional microhardness, or mineral loss measured by traverse microradiography). The analysis was performed using the software Review Manager (RevMan Version 5.3, The Nordic Cochrane Centre, The Cochrane Collaboration, Copenhagen, Denmark, 2014) and the results were graphically presented by means of forest plots.

## 3. Results

### 3.1. Search and Selection

The initial literature search identified 946 potentially eligible publications up to 31st December 2020 (91 articles in PubMed, 728 articles in Scopus, and 127 articles in Web of Science). One hundred and eighty-three duplicated records were removed. After the screening of the titles and abstracts, the researchers excluded articles that did not study the effect of treatment combining a 9300 nm $CO_2$ laser and fluoride on the prevention of caries, along with papers that were literature reviews, case reports, conference papers, and book sections. Full-text papers were obtained for the remaining 58 publications. A search of the references of the selected studies did not identify any additional publications that met the inclusion criteria. Fifty-three studies were excluded according to the exclusion criteria. The remaining five studies were found to meet the eligibility criteria and were included in this systematic review. All of them were in vitro studies. Figure 1 is a flowchart of the selection process for studies according to the PRISMA statement on literature searches [32].

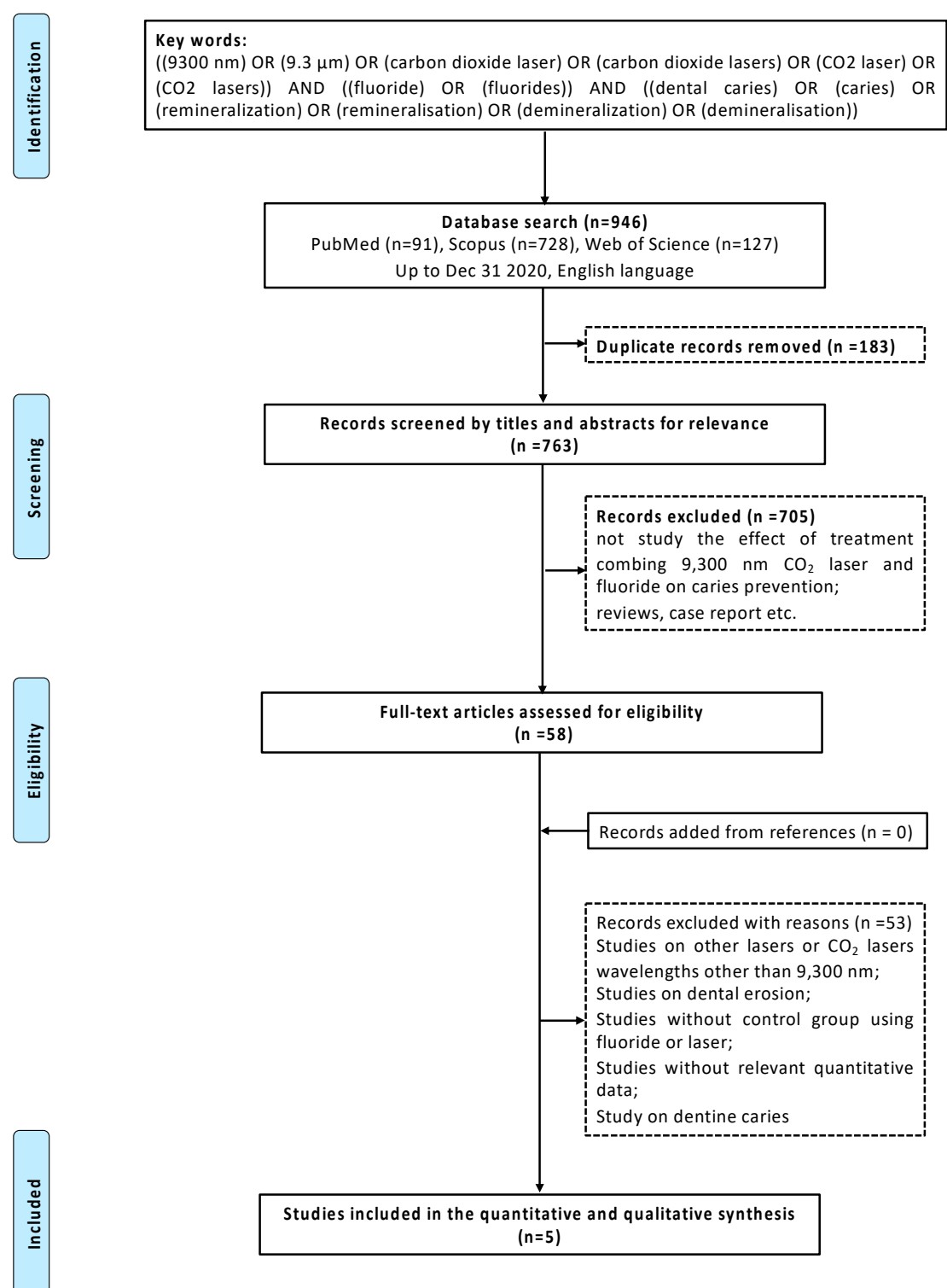

**Figure 1.** A flowchart diagram of the study selection process, according to the PRISMA statement.

*3.2. Descriptive Analysis*

Table 1 shows laser parameters used in the included studies in the systematic review. All the $CO_2$ lasers emitted short pulses in the microsecond (μs) range, and the fluences of most of them were below 10 J/cm$^2$. Table 2 shows descriptive extracted data in terms of study design from the included studies in the systematic review.

**Table 1.** Parameters for the 9300 nm carbon dioxide laser used in the studies of the systematic review.

| Authors, Year [Reference] | Laser Machine; Manufacturer | Cooling | Fluence | Pulse Duration | Beam Diameter | Frequency | Spot/ Motion | Irradiation Time |
|---|---|---|---|---|---|---|---|---|
| | | | J/cm$^2$ | μs | μm | Hz | mm/s | s |
| Can et al., 2008 [37] | Impact 2500; GSI Lumonics | Air and water | 20 | 15 | 320 | 100 | Motion 1.5 | - |
| Hsu et al., 2008 [38] | Impact 2500; GSI Lumonics | No | 1 | 15 | 100 | 200 | Motion 3.0 | - |
| Rechmann et al., 2016 [17] | Impact 2500; GSI Lumonics | No No | 4.6 5.12 | 5 6 | 250 250 | 43 43 | Spot Spot | 120 120 |
| Rechmann et al., 2020 [39] | Solea, Convergent Dental | Air Air | 1.4 1.9 | 11.4 14.6 | 630 630 | 100 100 | Spot Spot | 25 25 |
| Badreddine et al., 2020 [40] | Solea, Convergent Dental | Air | 0.6 0.8 1.0 | 17 22 27 | 1000 1000 1000 | 750 | Spot | 0.3 |

**Table 2.** Study design of the included studies in the systematic review.

| Authors Year [Reference] | Specimen | Study Design | Cariogenic Challenge | Fluoride, Concentration | Fluoride Application | Assessment |
|---|---|---|---|---|---|---|
| Can et al., 2008 [37] | Bovine enamel | In vitro, chemical model | pH 4.9, 9 days | Acidulated phosphate fluoride, 12,300 ppm | After laser irradiation | PS-OCT, PLM, TMR |
| Hsu et al., 2008 [38] | Bovine and human enamel | In vitro, chemical model | pH 4.9, 7 days | Acidulated phosphate fluoride, 12,300 ppm | After laser irradiation | PS-OCT, PLM, TMR, LIF |
| Rechmann et al., 2016 [17] | Human enamel | In vitro, chemical model | pH-cycling pH 4.4/7.0, 9 days | Sodium fluoride, 825 ppm | After laser irradiation | MHT |
| Rechmann et al., 2020 [39] | Human enamel | In vitro, chemical model | pH-cycling pH 4.4/7.0, 9 days | Sodium fluoride, 825 ppm | After laser irradiation | MHT |
| Badreddine et al., 2020 [40] | Human enamel | In vitro, chemical model | pH-cycling pH 4.4/7.0, 9 days | Sodium fluoride, 825 ppm | After laser irradiation | MHT |

PS-OCT, polarization sensitive optical coherence tomography; PLM, polarized light microscopy; TMR, transverse microradiography; LIF, fluorescence loss measurements; MHT, microhardness testing.

All studies were published between 2008 and 2020, with three studies published after 2010 [17,39,40]. All of the studies included were in vitro laboratory studies, and conducted at the University of California in San Francisco, USA. Among the five studies, there were two studies using bovine enamel rather than human enamel [37,38]. A chemical cariogenic challenge model was applied above seven days and for up to twelve days in all studies. In this collection, the majority of the studies used 1.23% acidulated phosphate fluoride [37,38], and the remaining ones used toothpaste/deionized water slurry containing 0.0825% sodium fluoride through calculation [17,39,40]. Besides, all studies applied fluoride after laser treatment. Mineral loss was the only criterion used in all studies to evaluate the demineralization of caries. Two studies measured mineral loss by traverse microradiography [37,38], whereas the other studies measured by cross-sectional microhardness [17,39,40]. The data from quantitative microradiography analyses were comparable with the data from cross-sectional microhardness profiles [41]. Therefore, the pooled data of mineral loss was assessed in the meta-analysis.

*3.3. Meta-Analyses*

The results of the meta-analysis were separately assessed for comparing combined treatment (F + L) with single fluoride (F) and single laser (L) treatment. Each comparison

was conducted on five studies which resulted in 10 comparisons, as some studies presented more than one experimental group. Figure 2 summarizes the findings of the meta-analysis comparing the effect of treatment combining fluoride and laser (F + L) and treatment using only fluoride (F) on mineral loss. As shown in Figure 2, the overall standardized mean difference in the meta-analysis was −1.84 (95% CI: −2.30 to −1.39). This indicates that the mineral loss of combined treatment was significantly less than that of the single fluoride treatment ($p < 0.00001$). The heterogeneity was found to be moderate ($I^2 = 49\%$).

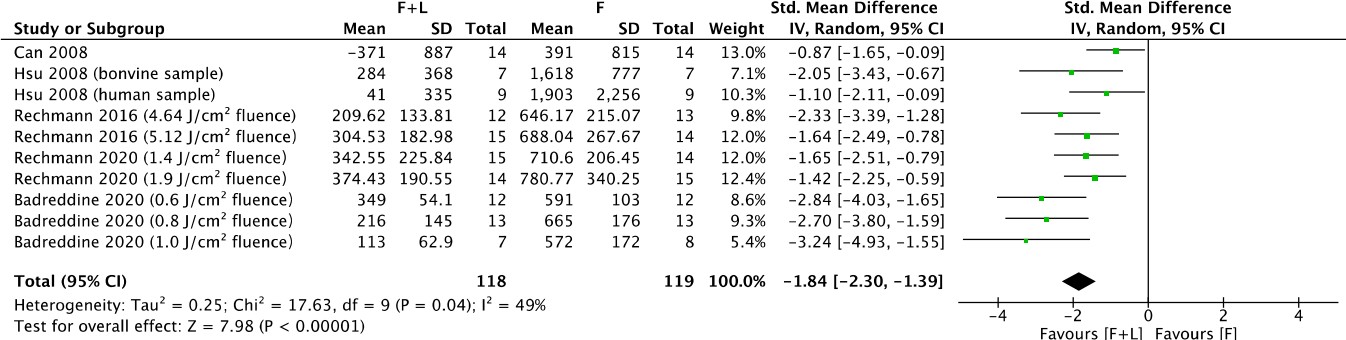

**Figure 2.** Meta-analysis comparing the effect of the treatment combining fluoride and laser (F + L) and treatment using only fluoride (F) on mineral loss.

Figure 3 summarizes the findings of the meta-analysis of the effect of the treatment combining fluoride and laser (F + L) versus single laser treatment (L) on mineral loss. Due to the high heterogeneity ($I^2 = 68\%$) of different studies, we performed a global analysis as well as a subgroup analysis. As shown in Figure 3, the overall standardized mean difference in the meta-analysis was −1.58 (95% CI: −2.13 to −1.03), indicating that the mineral loss of the combined treatment was significantly less than in the single laser treatment. Low heterogeneity ($I^2 = 0\%$) was observed for the subgroup in which mineral loss was assessed by transverse microradiography. The main heterogeneity comes from the subgroup in which mineral loss was assessed by cross-sectional microhardness, with $I^2 = 75\%$.

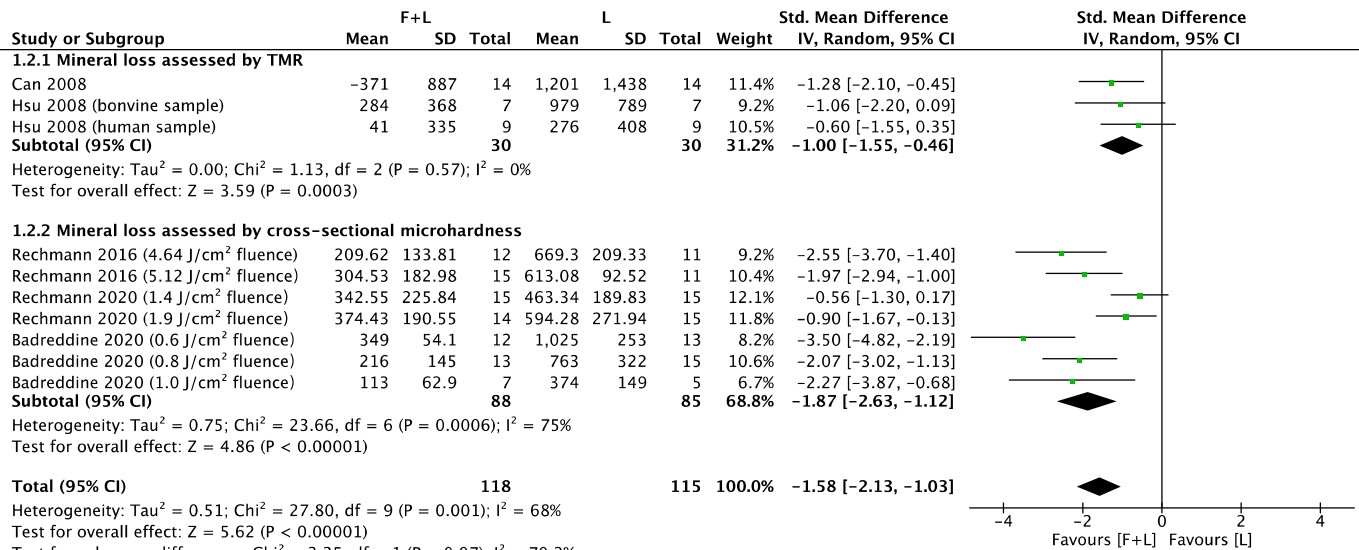

**Figure 3.** Meta-analysis comparing the effect of the treatment combining fluoride and laser (F + L) and treatment using only fluoride (L) on mineral loss.

### 3.4. Risk of Bias

Of the five studies included, two presented a medium risk of bias and three showed a high risk of bias. The results are described in Table 3.

**Table 3.** Risk of bias of the included studies in the systematic review.

| Authors, Year [Reference] | Can et al., 2008 [37] | Hsu et al., 2008 [38] | Rechmann et al., 2016 [17] | Rechmann et al., 2020 [39] | Badreddine et al., 2020 [40] |
|---|---|---|---|---|---|
| Quality check of samples | - | - | - | - | Yes |
| Randomization of samples | - | - | - | - | - |
| Sample-size calculation | - | - | Yes | Yes | Yes |
| Homogeneity of samples | Yes | Yes | - | - | - |
| Details of laser parameters used | Yes | Yes | Yes | Yes | Yes |
| Fluoride application protocol | - | Yes | Yes | Yes | Yes |
| Operator training | - | - | - | - | - |
| Blinding of operator | - | - | - | Yes | - |
| Risk of bias | High | High | High | Medium | Medium |

## 4. Discussion

This review aimed to justify the combined use of a 9300 nm $CO_2$ laser and fluoride to foster the prevention of enamel caries. It summarized the in vitro data of treatment combining fluoride and the laser versus the single use of the laser and the single use of fluoride. It showed that enamel treated with the laser and fluoride sustained less mineral loss against acid challenge than the enamel treated with the laser or fluoride only.

This study aimed to include original research on both enamel and dentine caries and both laboratory and clinical studies. However, no clinical or in vivo study was found after the initial search and screening. In addition, there was only one paper investigating the effect of the laser on dentine caries [42]. We did not want to draw a conclusion on dentine caries based on a single paper. Hence, we revised our objective and decided to focus on studies on enamel caries in this systematic review and meta-analysis.

This review was not registered in any web or institution. Registering the review prevents duplication and minimizes the opportunity for reporting bias. We searched the PROSPERO which provides a comprehensive listing of systematic reviews registered and found no similar review.

The progression of caries in dentine differs from that in enamel. Enamel caries occur mainly due to the dissolution of a mineral component as a consequence of bacterial acid attack [43], whereas both mineral dissolution and the proteolysis of the collagenous matrix are involved in dentine caries [44]. The study on dentine caries demonstrated that compared with L, F + L showed a decrease in lesion severity. However, the study found that compared with F, F + L did not show any statistical significance. Current scientific evidence of the effectiveness of the treatment combining a 9300 nm $CO_2$ laser and fluoride on the prevention of dentine caries is insufficient. It is important to carry out more research to investigate these issues.

The selected studies assessed mineral loss in different ways, such as through cross-sectional microhardness or traverse microradiography. Additionally, some studies included in the meta-analysis used bovine enamel rather than human enamel. Bovine enamel is an ideal alternative for studies on enamel caries. The collection of extracted human teeth is one of the major challenges in using human enamel specimens. In addition, variations in the age of the donors can give rise to variations in the mineral density of the enamel, which can affect the results of the study. In comparison, bovine enamel is more uniform in composition. However, bovine enamel is more porous, thus forming lesions more rapidly, displaying a rate of lesion formation three times faster than in human enamel [45]. However, using bovine and human enamel data in the quantitative discussion of artificial carious

lesion formation can be allowed, because the ratio of bovine enamel lesion formation rate in comparison to that in human enamel is almost constant [46]. For these reasons, we standardized the results of the studies by using a standardized mean difference rather than mean difference.

Irradiation with $CO_2$ laser can induce physiochemical changes in enamel such as hydrophobicity and zeta potential, which have been found to affect the adhesion of some bacterial species on enamel surfaces [47,48]. Moreover, it seems that $CO_2$ laser irradiation can directly impact the activity and viability of *Streptococcus mutans* biofilm on the surfaces of teeth [49]. The topical use of fluoride has been widely acknowledged for its antibacterial properties on cariogenic bacteria in the plaque [50]. The combined use of 5% sodium fluoride varnish and a 10,600 nm $CO_2$ laser was found to reduce bacterial numbers on dentine surfaces [51]. All the studies here used a chemical model rather than a microbiological caries model. Compared to the microbiological model, the chemical model used for the cariogenic challenge allows more control over the experiment conditions, which may generate fewer random errors in the results [52]. Three studies used simple mineralization models (with constant pH around 5) and the other two used pH-cycling models (pH 4.4/7.0). The latter models provide a periodic alternation of pH when sugars are metabolized to form a caries lesion [53]. A main limitation of the chemical models is that they completely ignore the microbiological aspect of the formation of caries [54]. To date, the effect of adjunctively using a 9300 nm $CO_2$ laser and fluoride on the action of cariogenic biofilms on hard dental tissues has not been demonstrated, and further research and exploration using a microbiological model are needed.

All of the included studies used a microsecond short-pulsed $CO_2$ laser. The duration of the pulse matches well with the thermal relaxation time of the enamel, which results in less thermal deposit and higher peak power. Only one study included used a $CO_2$ laser accompanied by water cooling [37]. Water cooling is important for controlling temperature. Enamel without water cooling manifested non-uniform surfaces with undesirable white asperities, which are unfavorable to acid resistance [55].

All of the studies included in this systematic review applied fluoride after laser treatment. Researchers found that fluoride application after laser treatment resulted in a significant increase in acid resistance of the enamel with fluoride application before laser irradiation. Fluoride application after laser irradiation produced a greater fluoride uptake in the enamel than fluoride application before laser irradiation [56]. However, there are some opposite findings. A few studies reported that applying fluoride prior to $CO_2$ laser treatment is more effective in preventing enamel demineralization against acid challenge [27,57,58].

A high heterogeneity was found in the meta-analysis comparing the effects on mineral loss of the treatment combining fluoride and laser (F + L) versus treatment only with laser (L). We performed a subgroup analysis to look for the source of the heterogeneity. To our knowledge, this grouping method is the best not only because it was determined according to the types of fluoride or methods of assessing mineral loss, but as it can also be regarded as classified according to research group. High heterogeneity was found in the subgroup assessed by cross-sectional microhardness. The high heterogeneity could be due to the sample size and variation in the laser parameters used in the study design.

In addition, all the studies in the subgroup in which mineral loss was assessed by cross-sectional microhardness received a treatment of 0.0825% sodium fluoride, and all the studies in the subgroup for which mineral loss was assessed by transverse microradiography received a treatment of 1.23% acidulated phosphate fluoride. Therefore, the subgroup in which the mineral loss was assessed by cross-sectional microhardness can also be regarded as the subgroup using 0.0825% sodium fluoride. For this reason, the results of the subgroup analysis suggested that the additional use of fluoride even in low concentrations still leads to significantly different results from the single use of laser.

In vitro conditions are different from in vivo conditions. The results of this review should be interpreted cautiously because in vitro studies have intrinsic limitations. In

addition, the parameters of the laser used in the included studies are diverse, which may influence the results in other dimensions. Besides, most of the included studies presented a high risk of bias. Well-designed clinical trials are necessary to verify whether the combined use of a 9300 nm carbon dioxide laser with fluoride will enhance the effect of caries prevention compared to the single application of each treatment.

## 5. Conclusions

In conclusion, the systematic review and meta-analysis suggested that irradiation with a 9300 nm $CO_2$ laser combined with the application of fluoride is superior to either $CO_2$ laser irradiation or fluoride application alone in preventing enamel caries in vitro. However, this review also found a high heterogeneity and a high risk of bias among the studies. Researchers should develop robust and cogent protocols for laboratory studies.

**Funding:** This research is funded by the National Natural Science Foundation of China (Project Number: 82001105).

**Institutional Review Board Statement:** Not applicable.

**Informed Consent Statement:** Not applicable.

**Data Availability Statement:** Not applicable.

**Conflicts of Interest:** The authors declare no conflict of interest.

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
