# Peer review of "Effects of Treatment Combining 9300 nm Carbon Dioxide Lasers and Fluoride on Prevention of Enamel Caries: A Systematic Review and Meta-Analysis"

_applsci, doi:10.3390/app11093996_

Round 1

Reviewer 1 Report

I read with interest the manuscript titled "Effects of treatment combining 9,300 nm carbon dioxide lasers and fluoride on prevention of enamel caries: A systematic re view and meta-analysis of in vitro studies", which well falls within the scope of Applied Science journal.
The authors conducted a systematic review and meta-analysis to investigate the influence combined application of 9300 nm CO2 lasers and fluorides (F+L), in comparison with single application of either CO2 lasers (L) or fluorides (F). The analysis showed that F+L was better than L and F, based on selected five studies where mineral loss was evaluated in vitro.

The abstract clearly summarizes the idea of the entire manuscript, and the reader can easily understand the effect of F+L on the mineral loss from the enamel. The analysis is clearly explained and following a current and accurate guidelines. The discussion and conclusions of the study are clear. 

This reviewer found some minor issues that could strengthen the manuscript, as follows;
1. In the Introduction section, please provide more specific rationale for analyzing the effect of the combined treatment as the topic to be investigated. The reason that "individual study cannot provide sufficient evidence to draw a conclusion", as stated by the authors, is not sufficiently specific as the rationale of this study.

2. Please specify the novelty of your work more clearly. Is this just "verification of pooled effect of data from previous studies"?

Reviewer 2 Report

This paper explored a systematic review and meta analysis on the effect of treatment combing 9300 nm carbon dioxide lasers and fluoride on prevention of enamel caries. There are a few comments as follows:

  1. In the introduction, you mentioned a few previous paper, whereas such as “there was no similar paper” was not mentioned.
  2. Has this study been registered in any web or institution, such as PROSPERO? If not, please state the reason or any similar paper was registered?
  3. As you mentioned, “The studies related to any types of studies (including in vitro, in vivo or clinical studies)” I understood only vitro studies were included finally, however all types of studies were considered. The title of this study is “in vitro studies”.
  4. Page 3 Line 116 the authors excluded “Studies without control groups using laser or fluoride” and “Literature reviews, case reports, conference papers and book sections”
. This makes me confused that did you included all types of papers or only intervention studies (when you mention control group)?
  5. Page 4 Line 146 “The risk of bias was classified according to the sum of “yes” received as follows: 1–3 = high, 4–5 = medium, 6–8 = low risk of bias.” Did you also classify the risk of bias followed by previous papers rather than following Cochrane handbook?
  6. Page 4 Line 157 “Statistical heterogeneity among studies was assessed using the Cochrane Q statistic and I2 test (>50% indicates significant heterogeneity)”. In the Cochrane handbook, 0-40%: considered not important; 30% to 60%: may represent moderate heterogeneity; 50% to 90%: may represent substantial heterogeneity; 75% to 100%: represents considerable heterogeneity. Please state I2 test in different levels.
  7. Page 7 Line 214 “The heterogeneity was found to be low (I2 =49%)”. This is low or moderate?
  8. In the first paragraph of discussion, the authors should summarize the findings of study.
  9. Page 8 Line 239-246 Should this paragraph move to the last part of introduction?

“Studies have indicated that laser treatment can work synergistically with the application of topical fluoride to further enhance its inhibitory effect.” References should be added.

  1. Page 8 Line 247-252 A few words such as however, besides, hence were competitive, which could be replaced by combining the sentences.
  2. Page 9 Line 291-296 “All the studies used a microsecond short-pulsed CO2 ” All the studies mean all the included studies?
  3. Page 9 Line 293 “Some studies used a CO2 laser accompanied by water cooling.” Please add references.
